

# Vegetation indices as proxies for spatio-temporal variations in water availability in the Rio Santa valley (Peruvian Andes)

Lorenz Hänchen[1], Cornelia Klein[2], Fabien Maussion[2], Wolfgang Gurgiser[2], Pierluigi Calanca[3], and Georg Wohlfahrt[1]

[1]Department of Ecology, University of Innsbruck, Austria
[2]Department of Atmospheric and Cryospheric Sciences, University of Innsbruck, Austria
[3]Agroscope Institute for Sustainability Sciences ISS, Zürich, Switzerland

**Correspondence:** Lorenz Hänchen (lorenz.haenchen@uibk.ac.at)

**Abstract.** In the semi-arid Peruvian Andes, the growing season is mostly determined by the timing of the onset and retreat of the wet season, to which annual crop yields are highly sensitive. Recently, local farmers in the Rio Santa basin (RSB) reported decreasing predictability of the onset of the rainy season and further challenges related to changes in rainfall characteristics. Previous studies based on time series of local rain gauges however, did not find any significant changes in either the timing or

intensity of the wet season. Both in-situ and satellite rainfall data for the region lack the necessary spatial resolution to capture the highly variable rainfall distribution typical for complex terrain, and are often questionable in terms of quality and temporal consistency. To date, there remains considerable uncertainty in the RSB regarding hydrological changes over the last decades.

    In this study, we overcome this limitation by exploiting satellite-derived information on vegetation greenness to reveal a robust and highly resolved picture of recent changes in rainfall and vegetation phenology across the region: As the semi-arid

climate causes water availability (i.e. precipitation) to be the key limiting factor for plant growth, patterns of precipitation occurrence and the seasonality of vegetation indices (VIs) are tightly coupled. Therefore, VIs can serve as an integrated proxy of rainfall. By combining MODIS Aqua and Terra VIs for 2000-2020 and several datasets of precipitation, we explore recent spatio-temporal changes in vegetation and water availability. Furthermore, we examine their links to El Niño Southern Oscillation (ENSO).

While different rainfall datasets tend to be incoherent in the period of observation, we find significant greening over the majority of the RSB domain in VI data, particularly pronounced during the dry season (Austral winter). This indicates an overall increase of plant available water over time. The rainy season onset and consequently the start of the growing season (SOS) exhibits high inter-annual variability and dominates the growing season length (LOS). The end of the growing season (EOS) is significantly delayed in the analysis which matches the observed dry-season greening. By partitioning the results into

periods of three stages of ENSO (neutral, Niño, Niña), we find an earlier SOS and an overall increased season length in years associated with El Niño. However, the appearance of Niño/Niña events during the analysed period cannot explain the observed greening and delayed EOS.

    While our study could not corroborate anecdotal evidence for recent changes in the SOS, we confirm that the SOS is highly variable and conclude that rainfed farming in the RSB would profit from future efforts being directed towards improving

medium-range forecasts of the rainy season onset.



# 1  Introduction

The Rio Santa valley in the tropical Peruvian Andes is characterized by high seasonal variability of precipitation with a rainy
season lasting from approximately September to April where 70 - 80 % of annual precipitation occurs followed by a dry season
with little to no rainfall (e.g. Schauwecker et al., 2014). In this region, rainfall seasonality is strongly controlled by tropical
easterlies related to the South American monsoon system (Garreaud, 2009). Interannual differences in rainfall totals may reach
up to 100 %, linked to high variability in the driving atmospheric circulation patterns. This variability is partly driven by the
El Niño Southern Oscillation (ENSO) phenomenon. However, ENSO influences on rainfall patterns in the tropical Andes are
complex and not coherent in space and time. For the Cordillera Blanca (the eastern mountain range of the Rio Santa valley),
studies suggest a general dry (wet) signal following El Niño (La Niña) events (Vuille et al., 2008; Maussion et al., 2015).
But this linear relationship does not hold true for all years/events and is dependent on individual, localised anomalies in the
upper tropospheric flows. Furthermore, the primary focus of most studies has been ENSO effects on anomalies in glacier mass
balances in the highest altitudes of the Cordillera Blanca, which might not reflect the effects across the Rio Santa basin (RSB)
at lower altitudes. At the same time, studies focussing on the Pacific watershed of the Peruvian coast suggest a complex pattern
where both dry and wet anomalies might occur in the adjacent mountain ranges (where the RSB is located) following El Niño
events (Sanabria et al., 2018, 2019; Rau et al., 2017).

Small-scale farmers living along the slopes of the RSB are cultivating their crops closely linked to the onset and retreat of the
rainy season. These subsistence-based cultivation practices are increasingly threatened by rural exodus, expansion of mining
activities, industrialization of agriculture and overall economic growth and modernisation (Carey et al., 2014; Crabtree, 2002).
Apart from these challenges local farmers recently reported perceived changes in rainfall patterns, which additionally threaten
their livelihoods (Mark et al., 2010; Perez et al., 2010; Gurgiser et al., 2016). Particularly, they reported a) a higher variability
in the onset of the rainy season which complicates the planning for an ideal sowing date, b) a higher occurrence of dry spells
during the growing season leading to dried up crops and c) more frequent occurrences of crop damaging events such as intense
rainfalls, hail and ground frost. In contrast to these reports, the same authors (Gurgiser et al., 2016) could not find evidence for
the reported patterns by analysing two local rain gauge time series.

The complex terrain of the Andes is an important factor hampering the robustness of information on rainfall patterns and
changes. Large-scale rainfall drivers like ENSO are modulated by topography over short distances, creating microclimates.
Hence, data from rain gauges, often of questionable quality, additionally suffer from insufficient spatial coverage. Therefore,
spatio-temporal distributions of rainfall across the valley and potential recent changes in patterns or seasonality still remain
uncertain and have been reported neither for the spatial domain of the RSB (Schauwecker et al., 2014; Gurgiser et al., 2016)
nor for larger scales (i.e. the tropical Andes region) (Vuille et al., 2003). But together with other climate variables, fine-scale
precipitation patterns are a dominant driver for changes in ecosystem productivity (Nemani et al., 2003; Huxman et al., 2004;
Knapp and Smith, 2001; Bonan, 2008; Beer et al., 2010; de Jong et al., 2013), and are of importance for downstream water
shortages which to date are only assessed by quantifying the glacier mass balance - runoff relation (e.g. Baraer et al., 2012;



Bury et al., 2013; Mark et al., 2010; Condom et al., 2012; Kaser et al., 2003) or by future projections with locally highly uncertain results (Urrutia and Vuille, 2009; Buytaert and De Bièvre, 2012).

In the particular climatic setting of the semi-arid Peruvian Andes, the growing season of vegetation is mostly determined by the onset and retreat of the rainy season. Given most agricultural land is rainfed, crops and managed grasslands similarly rely on the seasonal rains (Rodriguez-Iturbe et al., 1999; Svoray and Karnieli, 2011; Schwinning et al., 2004; Forzieri et al., 2014). Other potentially limiting variables (i.e. radiation and temperature) are of minor importance for ecosystem productivity or successful rain-fed farming at the transitions between dry and wet season (Camberlin et al., 2007). Therefore, a strong relationship of remotely sensed Vegetation indices (VIs), such as the Normalized Difference Vegetation Index (NDVI) (Rouse et al., 1974) or the Enhanced Vegetation Index (EVI) (Liu and Huete, 1995) can be expected to show a clear, albeit lagged response to rainfall (Richard and Poccard, 1998; Potter and Brooks, 1998; Wu et al., 2015). VIs have been successfully used for detecting climate anomalies (Karnieli et al., 2010), revealing long-term changes in climate (Richardson et al., 2013, 2018; Zhang, 2005) and understanding local effects of large-scale patterns such as ENSO (Kogan, 2000). VIs can also be exploited to calculate metrics of land surface phenology (LSP). Widely used metrics are related to phenophases, greening or senescence of plants, usually named start, peak and end of the season (SOS, POS, EOS) and can be used to deduce interannual variability or spatio-temporal changes in ecosystem status (e.g. Vrieling et al., 2013; Xu et al., 2016)

In this study, the overachieving goal is to shed light on the interannual variability and decadal changes of water availability in the RSB in the context of perceived changes by local farmers which do not match time series of rain-gauge data. Specifically, we aim to:

1. demonstrate that VIs can serve as useful proxies in regions with lack of high resolute climate data to infer seasonal rainfall characteristics and changes on high spatio-temporal resolution in semi-arid mountainous areas.

2. understand the robustness of the rainfall - VI relation in the particular setting of the RSB.

3. acquire insights on driving patterns of interannual variability of these variables by larger-scale circulation patterns.

## 2   Material and methods

### 2.1   Study area and local climate

The Rio Santa basin (also: Callejón de Huaylas) is located in northwest central Peru, approximately 400km northwest from the capital Lima (see Fig 1). In several sections, the valley is densely populated while the majority of the land surface is used either for agriculture in the lower and extensive grazing in the higher altitudes. The complex interactions between the Andes topography, position of the Intertropical Convergence Zone (ITCZ), ENSO and the South American Monsoon system shape the precipitation gradient between the Amazon basin, which is among the rainiest places worldwide (Killeen et al., 2007; Espinoza et al., 2015), and the dry deserts along the pacific coast with close to zero precipitation (Rau et al., 2017). The RSB is located between those two extremes and consequently there is a precipitation gradient between the Cordillera Blanca range on the east



slope of the valley and the Cordillera Negra range on the west slope within a few kilometers distance as seen by rain gauge measurements and satellite precipitation retrievals in Fig. 1.

## 2.2 Vegetation Indices data

For this analysis we acquired complete time series of NDVI, EVI and PR (Pixel Reliability) layers of the MODIS Terra & Aqua satellites (i.e. products MOD13Q1 (Didan, 2015a) and MYD13Q1 (Didan, 2015b), respectively) using LP DAAC's

Application for Extracting and Exploring Analysis Ready Samples (AppEEARS) for a subset covering the RSB in NetCDF format. Both products contain images with a spatial resolution of 250 m and are composited from the best radiometric and geometric quality pixels (i.e. low clouds, low viewing zenith angle and highest values of NDVI/EVI) in a 16-day observation period. The composites in MOD13Q1 & MYD13Q1 are purposely phased eight days apart and use the same spatial grid, which allows combining them. Both MOD13Q1 and MYD13Q1 were filtered to only retain pixels with the MODLAND QA criteria

'VI produced with good quality' and 'VI produced but check other QA' and in a second step by removing low quality VI's ('Lowest Quality', 'Quality so low that it is not useful', 'L1B data faulty' and 'Not useful for any other reason/not processed'). In a third step, only pixels with 'low' and 'average' aerosol quantity were included and pixels where adjacent clouds, mixed clouds and/or possible shadows were detected, were removed from the data. Finally, the two filtered datasets were combined to cover February 2000 to October 2020 in 8-day temporal and 250m spatial resolution. The consistent dataset was exported into

a set of GeoTiff files for being processed with the Decomposition and Analysis of Time Series software (DATimeS) software (Belda et al., 2020). All VI analyses shown in this study are based on NDVI, as EVI time series produced overall similar results and are therefore not presented.

## 2.3 Precipitation data

We used gridded precipitation data from The Climate Hazards InfraRed Precipitation with Station data (CHIRPS) dataset (Funk

et al., 2015) in $0.05° \times 0.05°$ spatial and 1-day temporal resolution and cut the data for the evaluated MODIS NDVI period 2000-2020 and the spatial extent of the RSB. CHIRPS rainfall is derived by a combination of satellite and rain gauge data. In particular, precipitation is derived from thermal infrared Cold Cloud Duration observations and blended with rain gauge data by weighted interpolation. Due to its comparably high spatial and very high temporal resolution it is suitable and regularly used for regional studies in complex terrain as found in the RSB (e.g. Rivera et al., 2018; Torres-Batlló and Martí-Cardona, 2020; Segura

et al., 2019). In addition, monthly L3 GPM - IMERG (Global precipitation measurement - Integrated MultisatellitE Retrievals) precipitation data were used for comparison (Huffman et al., 2012). IMERG provides global estimations of precipitation based on microwave satellite observations in combination with surface precipitation rain gauges. In contrast to CHIRPS, IMERG rainfall retrieval is closer to actual precipitation measurements, but suffers from coarser spatio-temporal resolution ($0.1°$ x $0.1°$). Finally, we compared our results with data from local weather stations operated by the National Meteorological and

Hydrological Service of Peru (SENAHMI). Stations that suffered from larger data gaps over the NDVI time period were excluded from further analysis. This resulted in three suitable stations, all located along the valley floor (see Fig. 3d for approximate locations).



## 2.4 VI time series pre-processing

As our study area covers a variety of land cover types, we used two state-of-the-art methods to derive LSP metrics from
the NDVI time series: 1. Whittaker smoother (wt) (Atzberger and Eilers, 2011) and 2. Gaussian process regression (GPR)
(Rasmussen, 2004), both implemented in DATimeS software (Belda et al., 2020). The Whittaker smoother (Whittaker, 1922)
calculates least squares with a penalty based on how noisy the input time series is. The smoothness is controlled by a single
parameter ($\lambda$). It is widely used for the preparation of remote sensing data before extracting LSP. GPR is a non-parametric
Bayesian approach where (hyper-)parameters are determined in a probabilistic way in the calculation. Recent studies were able
to show advantages of GPR over standard models for gap-filling and fusion of various biophysical parameters (Belda et al.,
2020; Pipia et al., 2019; Mateo-Sanchis et al., 2018). Besides being promising in terms of yielding accurate estimates, GPR
is different from other models since it determines uncertainty estimates for each pixel in addition to the fitted data. However,
as differences between the results of the two methods turned out to be negligible, all analyses shown are based on GPR. The
DATimeS software was set up by using a daily time step. To account for possible greening or browning trends in the NDVI
time series we used a seasonal amplitude of 30% to determine SOS and EOS. All other settings were DATimeS default settings
(i.e. Min. Prominence of 20%, Minimum Separation of SOS and EOS of 100 days and no further smoothing method applied).
Additionally we filtered the LSP results by masking the data on conditions: i) pixels where LOS was longer than 365 days,
ii) pixels where the amplitude or the maximum NDVI was one or greater, iii) pixels where the order of SOS, POS, EOS was
not given in any way (e.g. POS after EOS), iv) pixels where SOS, POS or EOS was 365 days after the first September of each
season were removed. Finally, we removed the upper and lower 1% percentile of SOS, POS and EOS to remove outliers. Parts
of our study area also contain areas with irrigated agriculture, where two (or more) maxima in the NDVI signal per season
are expected. To exclude such pixels where the seasonality is evidently decoupled from precipitation variability, we used an
approach based on autocorrelation analysis (Verstraete et al., 2008). The time series was split into 14 months (growing season
$\pm$ 1 month) for each season (e.g. 2000-07-01 to 2001-08-31) and a 3-weeks rolling window was applied to the calculation
of autocorrelation for each pixel and season independently. By detecting the local maxima of the autocorrelation, the number
of peaks in each pixel was detected and finally all pixels with more than one local autocorrelation maximum were excluded
from further analysis (on average 7.23 % pixels per season removed with $\sigma$=1.42%). These pixels are exclusively located at
the highest altitudes and close to the Rio Santa river. Additionally, we masked the whole time series with the global Land
Cover product of the Copernicus Climate Change Service (C3S) at 300m resolution (https://cds.climate.copernicus.eu/cdsapp#
!/dataset/satellite-land-cover?tab=overview, accessed June 2020). Specifically, we removed all pixels which intersected with
nine specific land-cover classes corresponding to flooded vegetation, urban areas, bare areas, water, snow and ice. However,
we did not account for land-cover changes during the 20-year time period and masked the whole timeseries with the ESA CCI
LC data from 2018.





## 2.5 Lag correlation

To compare the rainfall data (i.e. CHIRPS) with the VI time series, a cross correlation function was implemented yielding the best lag (in days) which refers to the highest pearson r for each pixel. Since the VI data has a resolution of approximately 250 x 250m and the gridded CHIRPS data of $0.05° \times 0.05°$, we compared each VI pixel with the CHIRPS pixel intersecting it by using a nearest neighbor approach. Best lag values and corresponding Pearson correlation coefficient and p-value were saved.

## 2.6 Onset and Retreat of the Rainy Season

Due to potential quality issues in the available rainfall data we evaluated if the timing of the LSP metrics (i.e. SOS and EOS) can be modeled from rainfall data. Therefore, we developed a simple soil moisture "bucket" model, to account for the temporal integrative nature of VIs in comparison to direct rainfall. The model simulates soil water content by daily consecutive balancing of input (rainfall) and output (evapotranspiration):

$$SWC(t) = SWC(t-1) + P(t) - ET(t) \tag{1}$$

where $SWC$ $[m^3/m^3]$ is soil water content, $P$ $[mm/day]$ is precipitation and $ET$ $[mm/day]$ is evapotranspiration at times $t$ $[day]$. As input, we used daily CHIRPS rainfall data (domain mean), while daily evapotranspiration was constant based on an estimation (at $2$ $mm/day$). The model starts at an initial value of SWC ($0.15$ $m^3/m^3$) and does not allow to completely evaporate the water in the bucket, once a defined minimum value ($0.05$ $m^3/m^3$) is reached, no further drawdown occurs. Similarly, a maximum value ($0.5$ $m^3/m^3$) that SWC can reach is defined. Simulated SOS occurs once a defined critical SWC value ($0.2$ $m^3/m^3$) is reached for the first time after reaching the minimum SWC. Simulated EOS occurs if the critical SWC for SOS and maximum SWC were reached before and at the time where SWC falls below a defined value of SWC ($0.35$ $m^3/m^3$).

Additionally, we tested threshold methods as published in (Gurgiser et al., 2016) and a method according to which seasonal rainfall is accumulated against the seasonal average (Liebmann and Marengo, 2001). However, these methods did not sufficiently correlate with the LSP metrics, at least for the threshold methods this might be due to a different contextual focus, as these metrics are closer linked to human perceptions. Further details are presented in Appendix A.

## 3 Results

### 3.1 Seasonal Relationship between NDVI and Rainfall

We first evaluated how gridded rainfall information from CHIRPS and MODIS NDVI relate on a seasonal basis. Figure 2a illustrates that the domain-average time series of NDVI shows lagged co-variability with the rainfall cycle, including a response to drier and wetter years. In-situ measurements indicate that seasonal rainfall shows a west-east gradient across the valley (c.f. Fig. 1). This is confirmed by the gridded datasets in Fig. 2b, where both CHIRPS and MODIS represent this difference in seasonal water availability between the two ranges. Monthly rainfall differences show less rainfall for the Cordillera Negra,





particularly during the early rainy season with approximately 12% for September, October and November compared to the
average for the entire valley. (Fig. 2d). While CHIRPS suggests only minor differences between the ranges during the peak
rainy season (Jan-Mar), corresponding lagged NDVI values (approximately Feb-Apr) remain lower (higher) on the Negra
(Blanca) ranges with a minimum in April (Fig. 2c). This illustrates that NDVI is a useful metric to capture the response of
vegetation to cumulative water availability in this region, which may better reflect vegetation and crop sensitivities than rainfall
metrics alone.

## 3.2 Decadal Changes in NDVI and Rainfall

We investigate potential rainfall changes in the RSB based on three different rainfall datasets and in comparison to vegetation
greenness as represented by the MODIS NDVI. While the rain-gauge dataset and CHIRPS (Fig. 3b and d, respectively) do
not show any changes in the observation period, the IMERG dataset (Fig. 3c) indicates a significant reduction of rainfall. In
contrast, the NDVI data (Fig. 3a) reveals a greening tendency across the RSB which indicates sufficient water availability and
thus, questions a decrease in precipitation sums.

On a sub-seasonal basis, monthly VI trends as shown in Fig. 4, reveal widespread greening particularly pronounced during
the dry season (approximately May to September). In May and August, this greening is in line with the CHIRPS data, while
the other dry season months show no clear signal. As illustrated in Fig. 2a, the NDVI signal is lagging behind the rainfall signal
and therefore correspondence between changes in rainfall and NDVI might also be affected by a few months lag (Tote et al.,
2011). Significant browning occurs in only a relatively small fraction of the area, consistently localized in urban areas or where
mines have been operating.

## 3.3 Characteristics of the Vegetation Growing Season

To further explore if and how the seasonality might have changed we extracted spatio-temporal information on phenology,
rainy season metrics and spatial lag information between NDVI and rainfall data. As Fig. 5 indicates, SOS and rainfall-based
simulated SOS show a large but corresponding interannual variability, while POS, EOS and rainfall-based simulated EOS show
smaller fluctuations over time. Consequently, the growing season length (LOS) is mostly governed by the high variability of
SOS. The Cordillera Negra shows both delayed SOS and earlier EOS in comparison to the Cordillera Blanca, while POS is
similarly distributed for both ranges, which agrees with the monthly differences shown in Fig. 2b-d). These differences between
Cordillera Blanca and Negra remain clearly visible even on the pixel scale (Fig. 6a-c) with a nine day later SOS, seven day
earlier SOS and 15 day shorter LOS (median values of all pixels). Neither SOS nor LOS show larger-scale changes over the 20-
year time series across the valley as seen in Figure 6. EOS on the other hand, is shifted towards later dates on the valley scale,
without dominant localised patterns that would suggest land-use change driving this shift (Fig. 6f). The simulated SOS from the
CHIRPS rainfall data using the bucket model ($\xi$ in Fig. 5) agrees with the LSP derived from the NDVI data ($r^2 = 0.72$, RSME =
3.0). However, the correlation of EOS is weak ($r^2 = 0.08$) but as variability in EOS is small, the simulated and the LSP-derived
EOS show high correspondence (RSME = 7.2, see also Figure A1). From the cross-correlation between NDVI and CHIRPS
rainfall data ("best lag", Fig. 6d), we find that over all years and all significant pixels the median lag is exactly one month (30



days) with an interquartile range of 18 days. However, similar to the LSP results, the lag variability is governed by the early season signals of NDVI and precipitation, while there is little variability in the late season where rainfall decreases linearly from peak monsoon to the start of the dry season. In consequence, lower lags are an indicator for higher early season water availability. Spatially, a similar pattern occurs where best lags correlate with morphology: Concave land surfaces generally show smaller lags of approximately 14-21 days in comparison to convex surfaces, which might be related to higher water availability after the initial rainfalls through rapid runoff along stream and river channels in these locations. The observed changes towards smaller lags in some locations (Fig. 6h) are most likely related to irrigation at lower altitudes and to changes in species composition, water availability and succession patterns in the highest altitudes.

## 3.4  Influence of ENSO on interannual Variability

Our results on vegetation greening seasonality reveal changes towards a later retreat of the rainy season, but drivers of the large interannual variability in SOS and rainfall patterns need to be understood. Since ENSO is a major driver of interannual variability across the region, we now want to evaluate its potential role for variability in LSP and whether it might explain the identified tendency towards later EOS over the last two decades. Therefore, we categorized NDVI and rainfall mean seasonal cycles by partitioning them into Niño, Niña and Neutral phases. As Fig.7b-d indicates, we find that early season (Oct, Nov, Dec) precipitation tends to be enhanced under El Niño conditions, favoring early greening after the dry season (Fig. 7a). However, at the same time, we find lower (higher) mean seasonal precipitation (September to August) under El Niño (La Niña) conditions. For the three investigated rainfall products these are -3.6, -7.9 and -1.5 % during El Niño phases (+7.1, +8.3, +4.4 % during La Niña) for CHIRPS, IMERG and local weather station observations compared to the mean seasonal precipitation of the complete time series. Regarding vegetation, the increased early season precipitation seems to trigger a contrasting greening response over the seasonal cycle in the RSB: An early onset of the rainy season is favourable for early plant growth and may allow earlier sowing for farmers. At the same time, reduced total rainfall sums seem to restrict greening later in the season. Expected reduced precipitation during El Niño is insignificant in our analysis, but mostly affects peak monsoon rainfall (see Fig. 7) when plant water stress should be low. This may explain why the associated reduction in mean annual precipitation and peak monsoon precipitation has little effect on the NDVI signal later in the rainy season (Fig. 7a). Although the investigated time series features only one multi-year El Niño event (seasons 2014/15, 2015/16), we suspect that the accumulated lack of rain during such events (-8.5, -3.9, +7.4 % and -20.7,-29.3, -17.49 % for the seasons 2014/2015 and 2015/2016 for CHIRPS, IMERG and local weather station observations) may have a cumulative detrimental effect on plant growth and result in overall browning tendencies. For the growing season 2016/2017 after the 2015/2016 El Niño this might be the case, as the growing season is clearly delayed (for most pixels the latest SOS, POS and EOS of the whole time series, compare Fig. 5). However, for this particular growing season, November rainfall is extremely reduced (by -52, -64, -92 % for the three rainfall products), which might be unrelated to the previous El Niño event but a similar pattern occurs in the growing season 2005/2006, where SOS is severely delayed and November precipitation is reduced (by -48, -26, -85 % for the three rainfall products) following a phase of El Niño. As Fig. 8 shows, anomalies in NDVI show a non-linear response to anomalies of the Niño 3.4 index. From September to December though, there is a positive correlation between the two variables with 18 - 28% explained variance,





significant only for November and December. In May, where the strongest greening occurs (see Fig. 4), the correlation remains insignificant at approximately 7% explained variance, similar to other months at the end of the rainy season. Therefore, we cannot explain the observed changes in NDVI and EOS by ENSO alone.

## 4 Discussion

By combining metrics of LSP, rainy season onset/retreat and statistical analyses of spatio-temporal data of MODIS NDVI and rainfall, we aimed at creating a more robust picture of spatial differences in water availability across the RSB and of changes therein. The observed change towards vegetation greening, particularly widespread during the dry season, strongly suggests an increase in water availability, which is not captured by rainfall data. However, CHIRPS rainfall data appears to be adequately capturing rainy season variability as we were able to reproduce NDVI-based SOS and (to a lesser extent) EOS

with a simple rainfall bucket model (see Fig. 5). This is an important finding, as ground truth precipitation observations are rare and often of doubtful quality. As the strongest greening occurs during the dryer months when precipitation sums are very small, these changes might be below the precision of precipitation measurements, particularly in complex terrain. In contrast to the agreement between CHIRPS and MODIS NDVI data, our analysis of different rainfall datasets on domain scale gives inconclusive and inconsistent results regarding changes in annual precipitation totals, as previously reported by other authors

(e.g. Gurgiser et al., 2016; Schauwecker et al., 2014; Vuille et al., 2003). This illustrates the feeble precipitation data basis and the uncertainty that comes with assessments of rainfall variability in the region.

Overall, we found the mean seasonal cycle of NDVI across the RSB to be shifting towards higher values since 2000, with a reduction in amplitude linked to more pronounced late wet season / dry season greening. As many studies on changes in VIs in semi-arid areas suggest, greening patterns are not coherent and dominant drivers are diverse. Although currently greening

appears to be the dominant signal across the Andes (and many other regions), one has to account for regional climate and land-use from case to case (Fensholt et al., 2012). The same applies for studies beyond regional scales (i.e. Peru), where the diversity of ecosystems and gradients in environmental variables may constrain transferable conclusions (Polk et al., 2020). Previously, a variety of potential drivers for greening in the tropical Andes were reported. Among these are primary succession of recently deglaciated areas (Young et al., 2017), forestation activities (e.g. Aide et al., 2019) and agricultural land use expansion (Bury

et al., 2013). Although these mechanisms most likely also occurred in the Rio Santa basin during the observation period, they cannot explain how greening during the dry season occurs independent of altitude, aspect or land-cover type. By visually comparing pixels which show very intense values of greening with RGB-imagery, we discovered some areas which were affected by land-use change. These were mainly located in higher altitudes dominated by grassland/shrub (so-called Puna) ecosystems. In some of these locations, the changes were related to afforestation of evergreen conifers in certain locations of

the Cordillera Blanca (not shown). These pixels however only occur in small numbers and therefore cannot be the dominant cause of the identified greening. Hence we are confident that the widespread greening, particularly over the drier months, is linked to increased water availability indicating potential changes in the seasonality of rainfall and vegetation growth.



As known for the Amazon, ENSO-driven extreme events such as the drought during 2015/2016 can have complex effects such as having contrary anomalies in greening and photosynthesis for forests (Yang et al., 2018). For the (tropical) Andes region, little research was conducted regarding effects of larger scale circulation on vegetation. Related to farming, the highly variable SOS, and consequently LOS in the RSB is probably the largest challenge for farmers as planning for sowing and crop choice can be difficult under these conditions. This is especially pronounced on the Cordillera Negra where water availability is lower and LOS is shorter. The spatially widespread trend towards delayed EOS dates is in line with the identified dry season greening.

Influences from anthropogenic activities can potentially cause a decoupling of rainfall and vegetation. This can be related to land-use practices such as irrigation, fertilizing or tilling. However, there are several arguments for the validity of our analysis. First, large parts of the RSB are characterized by small-scale, subsistence based, rain-fed, non-industrial agriculture where a large-scale decoupling is not expected. Hence, we account for areas (i.e. at the valley floor) where a multi-modal growing season is realized by irrigation. Second, increasing glacial melt during the past decades might have increased (sub-surface) runoff and facilitated an extension of the agricultural growing season by irrigation. But as neither the magnitude nor dissemination of greening is distinguishable between the glaciated Blanca and non-glaciated Negra slopes and hence farmers predominantly reported negative impacts (c.f. Gurgiser et al., 2016) related to climate, this seems unlikely to be relevant. However, potential change in water availability during the late wet season remains not fully understood as i) the vegetation decouples from the rainfall signal with increasing rainfall sums, where the explanatory power of VIs can be limited and ii) the availability of cloud-free scenes is poor during the wet season which causes uncertainties (see dry and wet season panels in Fig. 4). However, the delay in EOS in a large proportion of the valley (see Fig. 6f) suggests a potential shift towards a slower retreat of the rainy season and/or slower decay of plant available water accumulated during the rainy season.

Taken together, we are mostly unable to confirm the local farmers reports (see Section 1). We cannot find an increasing variability in SOS over the period of observation, but as the interannual variability is generally very high, this perception is comprehensible if experienced simultaneously with challenges of other nature. Furthermore, the farmers reported increasing dry spells and more frequent occurrences of detrimental events (e.g. hail, frost). Regarding the dry spells, one methodological constraint is the focus on seasonal unimodal pixels, as severe drought events during the growing season might result in bi- or multi-modal VI seasonal cycles. However, a significant increase of severe dry spells is somewhat contrary to the observed greening pattern on regional scale but cannot be precluded on the local scale or might be not noticeable in the VI signal as farmers might take measures if their crops are threatened by droughts. Regarding extreme meteorological events, our analysis does not allow clear statements as the information in NDVI is accumulative and such events might occur very locally. But again, we do not expect detrimental effects on the seasonal cycle of the majority of the valley as we find widespread greening.

In recent years, our understanding of the hydroclimatological mechanisms improved, particularly in the Andean regions. However, the discovered changes of water availability are spatially variable across the Andes as different interacting mechanisms modify the hydroclimatic system on different timescales (e.g. ENSO (Garreaud, 2009; Arias et al., 2021), Pacific Decadal Oscillation (PDO) (Campozano et al., 2020), seasonality of Southern Pacific Anticyclone (al Fahad et al., 2020) and Bolivian High (Segura et al., 2019) circulation systems and consequently displacement of the ITCZ). No consistent pattern of rainfall in-



or decrease for the period 2000-2020 is reported for either the tropical Andes (Rabatel et al., 2013) or the RSB (Schauwecker et al., 2014; Gurgiser et al., 2016). Here, we find increased early season precipitation under El Niño conditions with only one

significant month and small rainfall sums. Hence, we find non-significant tendencies of decreased MAP values which are in line with glacier mass balance studies in the RSB (Kaser et al., 2003; Vuille et al., 2008; Maussion et al., 2015). However, the modulation of dry season precipitation is rarely the focus of neither glaciologists nor climatologists and therefore remains poorly understood.

Understanding the drivers of the greening in the RSB remains challenging and raises several questions. We found that ENSO

sequences for the observation period cannot explain the observed greening and delayed EOS. This is in line with a study on the impact of ENSO cycles on continental evaporation by (Miralles et al., 2014). They suggest that El Niño is associated with negative evaporation anomalies in parts of the Andes and illustrate a recovery from El Niño dominated evaporation conditions until approximately 2001 towards La Niña dominated conditions starting 2007. The early 2000s have a neutral to El Niño tendency though, which again suggests that ENSO phases are unlikely to be the dominant driver for the later EOS.

Globally, $CO_2$ fertilization is the dominant driver for vegetation greening (e.g. Sitch et al., 2015; Zhu et al., 2016) as photosynthesis rates are accelerated and water use efficiency of plants can be increased by stomatal closure with higher $CO_2$ availability. However, as water limitation can negate these benefits (e.g. Gray et al., 2016; Reich et al., 2014) and we find hints towards higher dry season rainfall we suggest the greening to be governed by higher water availability as previously observed for other, better investigated semi-arid climates (e.g. Sahel (Dardel et al., 2014; Brandt et al., 2019; Huber et al., 2011; Hickler

et al., 2005; Herrmann et al., 2005; Anyamba and Tucker, 2005; Eklundh and Olsson, 2003), Africa (Fensholt et al., 2012) or Australia (Donohue et al., 2009). The observed greening trend might also induce a feedback of increased transpiration bringing more moisture from the soils into the atmosphere which might be especially relevant during the dry season where this could lead to beneficial recycling of moisture and promote rainfall (Spracklen et al., 2012).

## 5 Conclusions

Changes in variability and amount of rainfall are great concerns for local society as many inhabitants of the RSB are subsistence-based farmers who rely on rain-fed agriculture. To date, drivers of changes in water availability in the RSB remain unclear and the feeble climate data basis hinders understanding spatial patterns and temporal trends.

Our study illustrates that VIs can be exploited as an integrative proxy of water availability and to examine the plausibility of rainfall data at regional scale and in data-scarce environments. Specifically, we quantified changes and variability of NDVI,

derived land surface phenology metrics and analyzed several rainfall products. We find changes in rainfall in between three products not to be coherent in space and time, while the VI data reveals a widespread greening trend, particularly pronounced during the dry season with low rainfall sums. The onset of the monsoon and consequently the growing season is strongly variable, while peak monsoon and the end of the wet season exhibit little variability in time. We find indications of increased early season but decreased peak monsoon precipitation during El Niño events, resulting in favorable conditions for early plant

growth as water availability is crucial during early season but less important during peak monsoon.

In consideration of the high variability in SOS and associated challenges for farmers, we suggest that future research should attempt to improve SOS forecasts derived from atmospheric circulation patterns. This could enable farmers to develop strategies to decrease risks of crop failure and optimize sowing dates. Although remote sensing nowadays provides information at unprecedented spatial resolution, we also emphasize the need for more and high quality local measurements (e.g. auto-
matic weather stations, flux measurements and LTER sites) to broaden the knowledge on the coupling between vegetation and hydroclimatic components in the Andes.

*Code and data availability.*   Code and datasets generated and/or analysed during this study are available from the corresponding author on reasonable request.

*Author contributions.*   L. Hänchen performed the analysis and wrote the paper. C. Klein, G. Wohlfahrt and F. Maussion adviced and assisted
L. Hänchen in the analysis. C. Klein prepared code for trend analysis, G. Wohlfahrt developed the bucket model and F. Maussion pre-processed the local weather station and rain-gauge data. W. Gurgiser provided valuable local expertise on the RSB. All authors contributed to the interpretation of the results and to the writing and/or review of the paper.

*Competing interests.*   The authors declare that they have no conflict of interest.

*Acknowledgements.*   This study was conducted in the frame of the AgroClim Huaraz project, funded by the Earth System Sciences Program
of the Austrian Academy of Sciences (OEAW). We thank Mario Rohrer for providing access to the METEODAT platform where we acquired SENAHMI weather station data. Many thanks to Santiago Belda for his support with the DATimeS software.

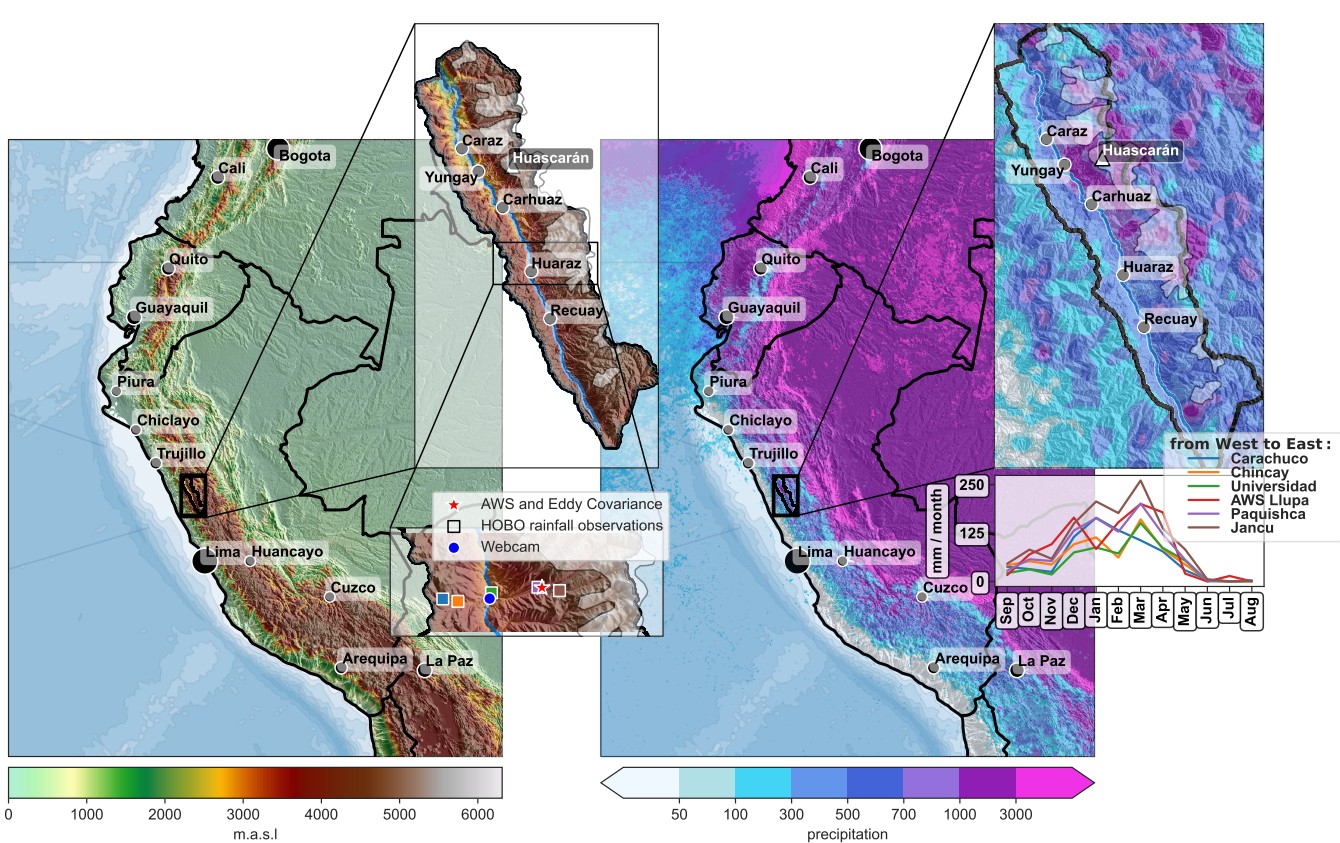

**Figure 1.** Left: Overview of the topography (based on SRTM data, Kautz (2017)). Important towns are shown (relative population by markersize). Black box marks RSB location and inset shows upper RSB including most important towns. Blue line indicates the Santa river, the range west (east) of the river is the Coordillera Negra (Blanca). Approximate glacier outlines are shown in white polygons. Small inset panel shows locations of rainfall observation transect by the AgroClimHuaraz (https://agroclim-huaraz.info/) project. Right: TRMM rainfall climatology (Bookhagen and Strecker, 2008) shows rainfall gradient over central-west South America and the RSB (Inset). Lower right panel illustrates the East-West precipitation gradient in the RSB of rain-gauge observations (from 2016 to 2019).





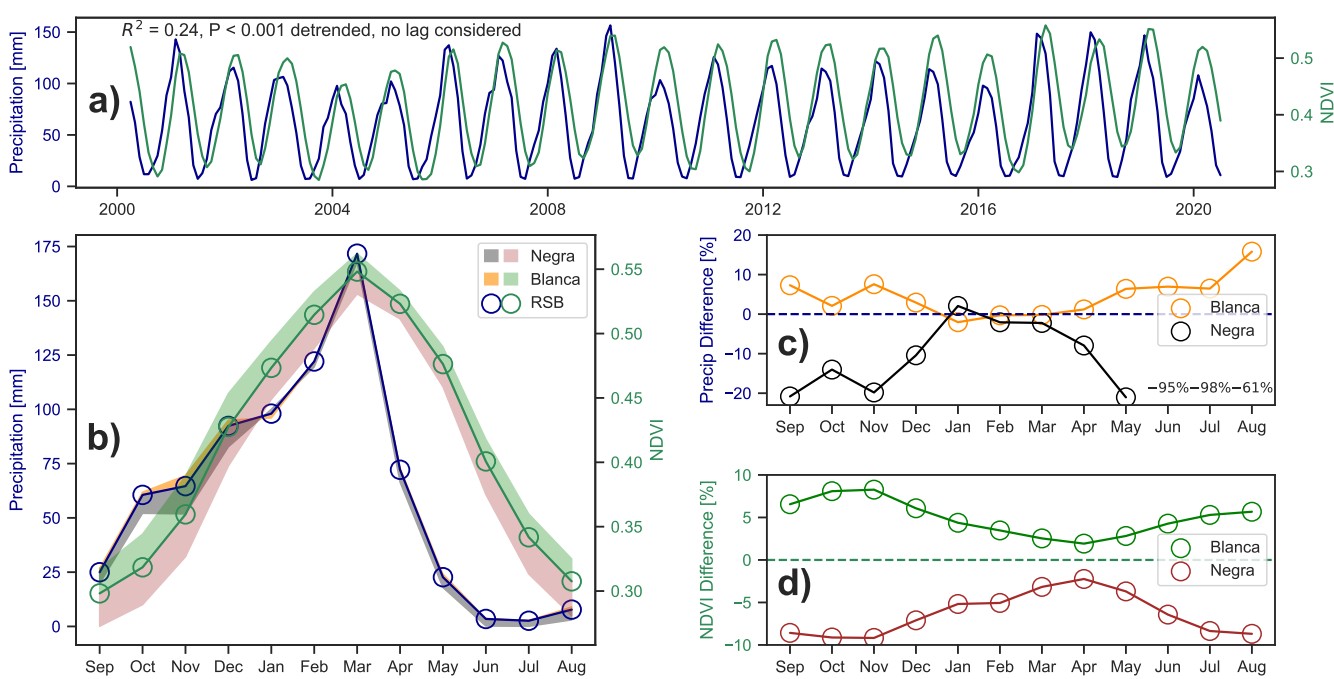

**Figure 2.** a) Domain mean cycles (4-month rolling mean) of NDVI and CHIRPS rainfall data between 2000 and 2020, b) Seasonal cycle of whole RSB and deviations of Cordillera Negra and Cordillera Blanca for CHIRPS and NDVI data. c) Relative differences in CHIRPS rainfall of Coordillera Negra and Coordillera Blanca against the domain mean seasonal cycle. d) same as c) but for MODIS NDVI data.

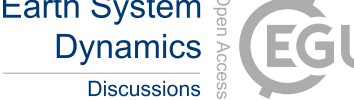

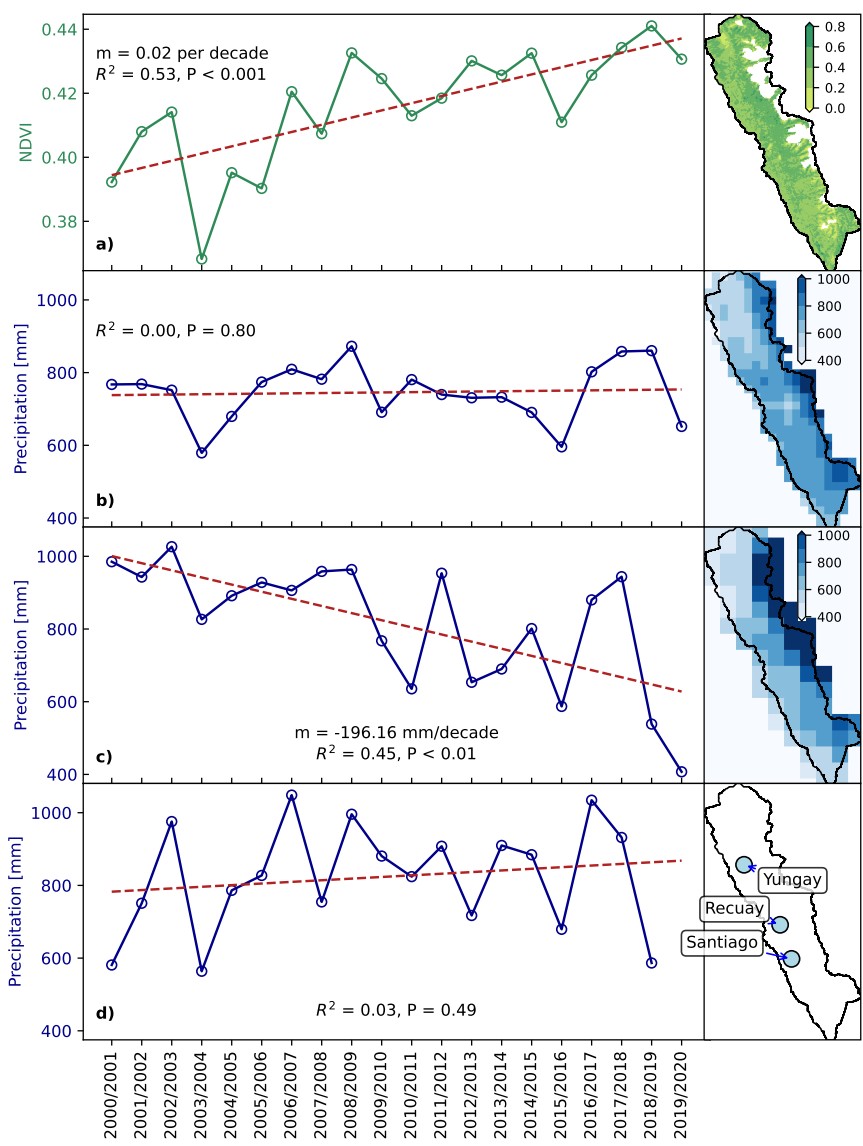

**Figure 3.** Seasonal domain mean time series and linear regressions for NDVI and three different rainfall products: a) MODIS NDVI, b) CHIRPS, c) IMERG and d) local weather station data (SENAHMI). Small maps show mean NDVI over the time series, mean annual precipitation sums and weather station locations, respectively.





**Figure 4.** Monthly greening and browning of NDVI. For months with at least 15% significant pixels, median slopes values ($\tilde{x}$) and interquartile ranges ($25 - 75\%$) are shown. Only significant pixels ($P < 0.05$) are shown, white color indicates non-significant pixels, while grey areas correspond to no-data. Pie charts show relative frequencies of greening, browning and non-significant pixels. Small panels show domain mean CHIRPS rainfall data for the respective month and additionally decadal slope (m) and linear regression statistics for significant ($P < 0.05$) relationships.





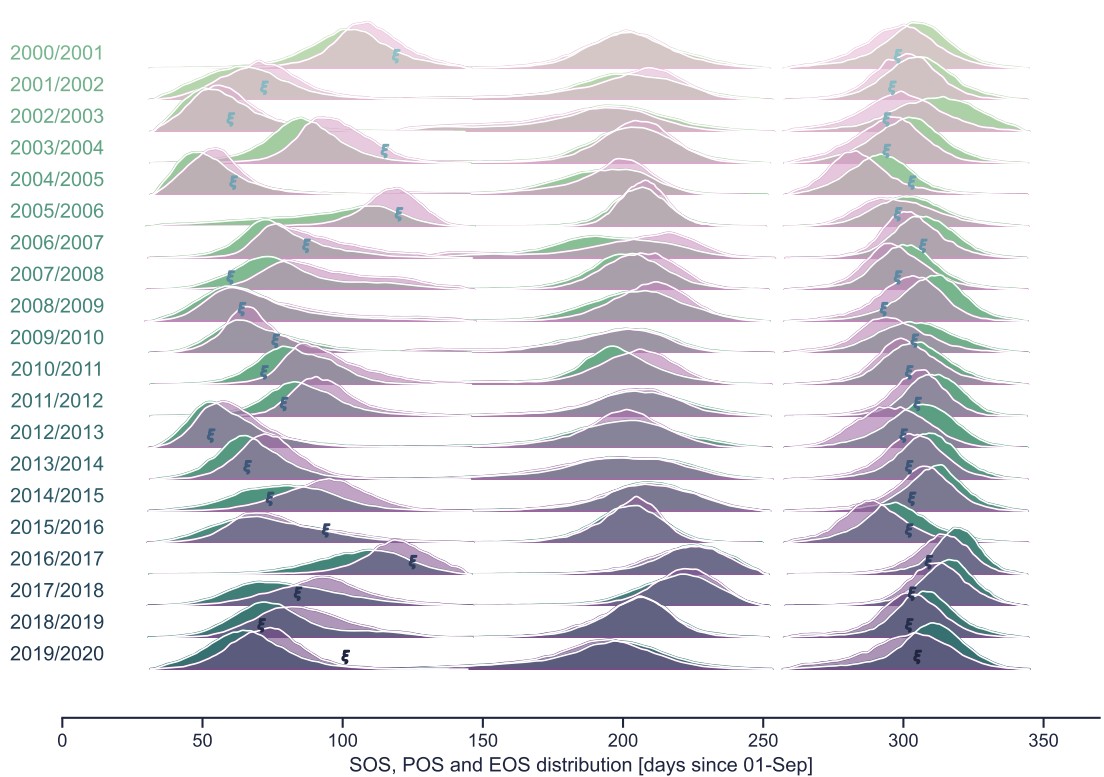

**Figure 5.** Kernel density estimations (KDEs) for SOS, POS and EOS for each growing season of all valid pixels in the RSB. Green (Purple) color refers to pixels located east (west) of the Rio Santa (Cordillera Negra and Blanca). Additionally, bucket-model simulated SOS and EOS ($\xi$) are shown for each season by using CHIRPS rainfall data averaged over the whole RSB domain.





**Figure 6.** Maps of LSP and NDVI-rainfall lag correlation analysis. First row: Median values of a) SOS, b) EOS, c) LOS, d) best lag between CHIRPS rainfall and MODIS NDVI for the RSB. Only pixels where the full time series is available (20 seasons) are shown. Second row: Linear regression for the same parameters, maps show decadal slope of the same parameters, inset scatter plots show time series of the domain median values with regression statistics if the regression is significant ($P < 0.05$). Only significant pixels ($P < 0.05$) are shown, white color indicates non-significant pixels while grey areas correspond to no-data. Pie charts indicate relative percentages of significant pixels, where red color indicates a forward shift and blue color a backward shift of the LSP metrics or increase/decrease in case of h) respectively.



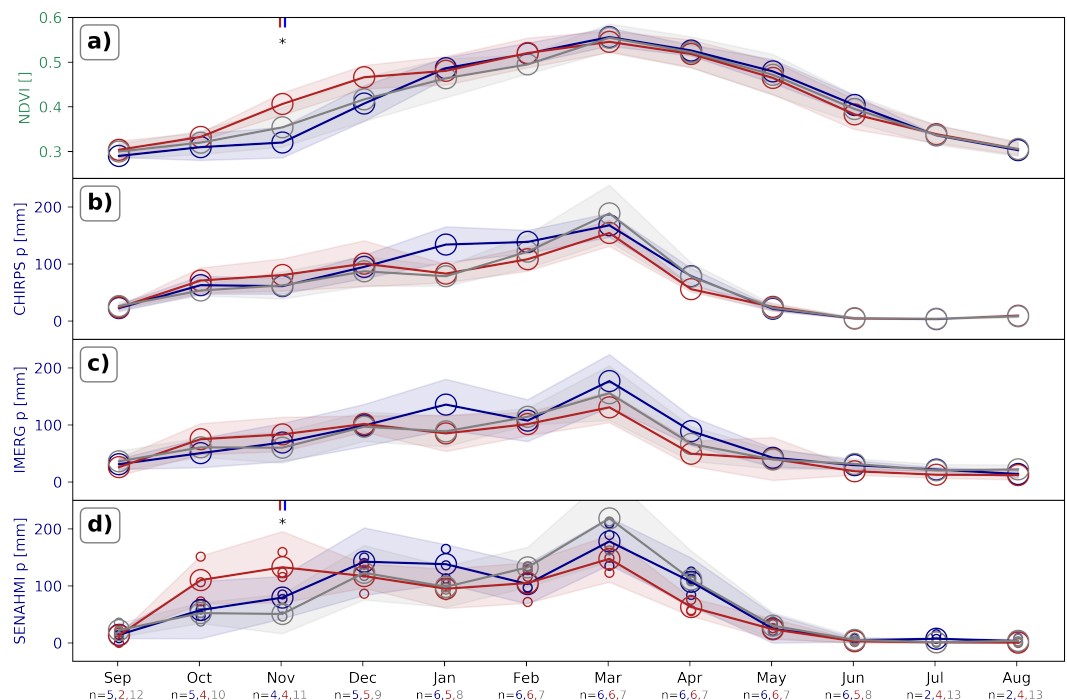

**Figure 7.** Mean monthly seasonal time series for 2000-2020 time series of NDVI and three rainfall products. Red (blue, grey) color indicates month in El Niño (La Niña, Neutral) classification after Trenberth (1997) of Niño 3.4. sea surface temperature anomalies (SSTa). We shifted the time series of SSTa by 3 month forward to account for lagged responses of rainfall in the RSB (Maussion et al., 2015). Below the x-axis, the number of months of each phase are displayed. Stars indicate significant results according to a Kruskal-Wallis and post hoc Conover's test ($P < 0.05$, corresponding phase marked by colored bars above the star). For panel d), the average time series of three stations in the RSB were used, smaller circles indicating values of the three individual stations. Locations of these stations are shown in the lower right panel of Fig.3.





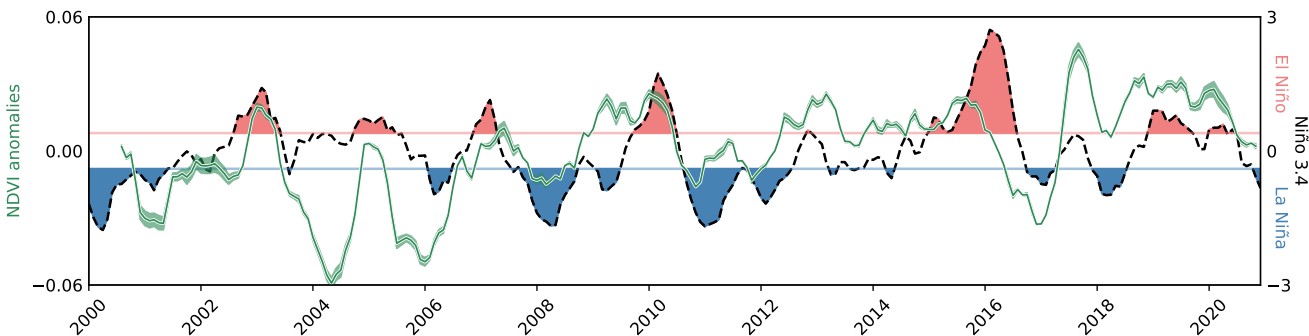

**Figure 8.** 7-month running average of monthly NDVI anomalies for the CdH domain and unsmoothed monthly 3-month shifted Niño 3.4 SSTa time series Niño/Niña events classified after Trenberth (1997) with a threshold value of ±0.4. Shaded areas represent $1\sigma$ of all valid pixels.



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





**Appendix A: Rainy season metrics**

To compare the timing of LSP indicators, we acquired metrics for the onset and end of the rainy season. We compared three methods methods, first we used the definition suggested by Gurgiser et al. (2016) where the day of the rainy season onset for each year is selected once three requirements are met simultaneously:

1. The day of the onset must have Precipitation > 0 mm

2. The sum of precipitation on the day of the onset and the following six days must be 10 mm or more.

3. The sum of days where precipitation occurred (> 0 mm) on the day of the onset and the following 30 days must be > 10 days.

Regarding requirement two, we additionally applied a threshold of 8 and 12 mm to test whether there might be higher agreement with LSP results. To determine the end of the rainy season (or the start of dry season respectively) the authors suggested two requirements to be fulfilled and as for the onset we applied additional thresholds of 8 and 12 mm for condition two.

1. The day of the end of the rainy season must have zero Precipitation ($P = 0mm$)

2. The sum of precipitation on the day of the end of the rainy season and the following 45 days must be less than 10 (8, 12) mm.

Second, we used a method successfully tested for the brazilian Amazon by Liebmann and Marengo (2001), where seasonal rainfall is accumulated against the seasonal average:

$$A(day) = \sum_{n=1}^{day} R(n) - \bar{R} \times day \tag{A1}$$

where $A$ is the cumulated rainfall anomaly of each day, $R(n)$ the daily rainfall of each particular season and $\bar{R}$ the seasonal mean daily rainfall. The particular days of onset and end of the rainy season are defined as the local minima and maxima of $A$ in the time series for each season.



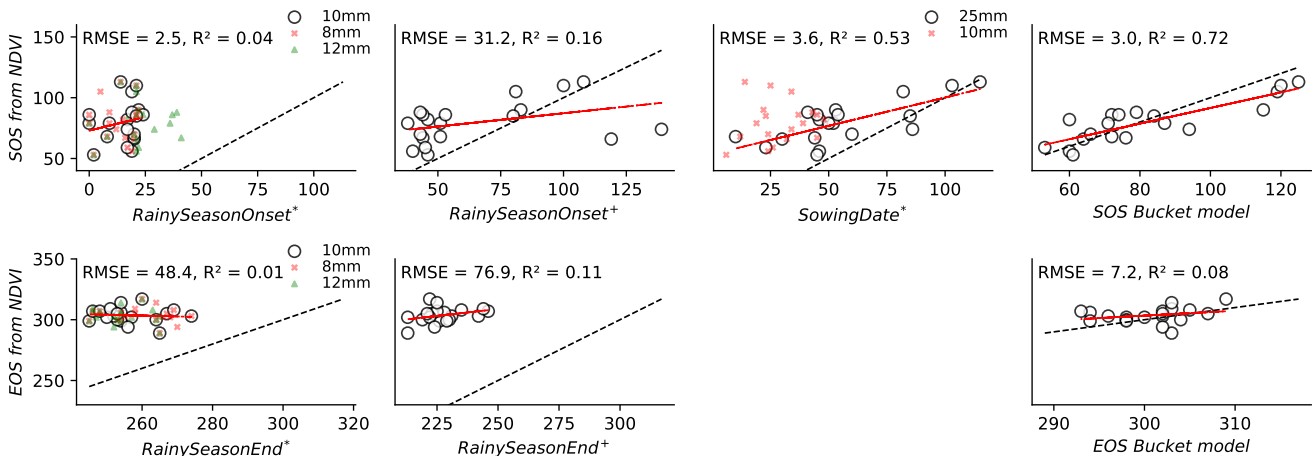

**Figure A1.** Scatterplots of calculated metrics against median SOS and EOS derived from LSP. * indicates metrics as published by Gurgiser et al. (2016), + by Liebmann and Marengo (2001) as described in Appendix A. Details on the bucket model in section 2.6. Regression lines relate to black circles in all plots.