# Peer review of "Widespread greening suggests increased dry-season plant water availability in the Rio Santa valley, Peruvian Andes"

_Earth System Dynamics, 2021_

## Author Comment (AC1)

**Review #1:**

*Overall, the manuscript is well written, and has several interesting findings. But my concern is that its title "Vegetation indices as proxies for spatio-temporal variations in water availability in the Rio Santa valley (Peruvian Andes)" is inadequate to what they are presenting on the paper. Based on the title, I expected the manuscript will be more focusing on the technical issues of how well satellite-based vegetation index captures spatiotemporal variations in water availability. However, the manuscript provides general characterizations about the relationships between vegetation index and precipitations, land surface phenology retrieval, and land surface phenology and larger-scale circulation patterns (i.e., ENSO). They also presented long-term greening and browsing without specific attributions of why. Therefore, I would recommend revising the title and relevant sections and expressions, especially for their overall goals.*

We thank the reviewer for the evaluation and the comments. We agree that the title is not ideally representing our work and we changed it to: "Timing and trends in vegetation greening indicate increasing plant available water in the Rio Santa basin (Peruvian Andes)". Regarding the reviewers comment that we "…*presented long-term greening and browsing without specific attributions of why*" we made amendments to the abstract and relevant text sections to specify more clearly that increased water availability is the key driver (and not e.g. land use change or $CO_2$ fertilization effects) of greening in the area. However, we additionally would like to point out to the reviewer that we addressed this issue thoroughly in the Discussion (i.e. lines 255 to 282 and 324 to 338). Additionally we reformulated in a more specific way our goals, adopting a 1:1 correspondence to the subsections in Section 3, "Results".

---

## Author Comment (AC2)

**Review #2:**

*In this analysis, the authors attempted to provide a new alternative to track the spatial-temporal variations in water availability in Rio Santa basin (RSB). They acknowledged the limitation of in situ and remote sensing-based rainfall datasets for complex terrain, i.e., the problematic quality and temporal consistence. While it is surprising that the authors found that satellite-derived vegetation greenness (also phenology) was coupled well with recent changes in rainfall (CHIRPS, a combination of satellite and rain gauge data). The authors proposed a "bucket" model to better fit vegetation phenology derived from MODIS NDVI, the concerns also raised in the criteria in extracting SOS/EOS. Some additional serious issues related to the methods and result interpretation, the organization of discussion would weaken the reliability and implication of this study. A major revision is therefore recommended.*

We thank the reviewer for the evaluation of our manuscript and the insightful comments.

*First, I have noticed that the authors claimed that both in situ and remote sensing precipitation datasets were questionable in signifying the changes in the timing and intensity of the wet season (e.g., lines: 4-7). However, it is quite strange that the authors used such dataset to demonstrate that the pattern of precipitation occurrence and the seasonality of vegetation indices are tightly coupled (e.g., lines: 10-11). Moreover, the authors used gridded precipitation data as a proxy of water availability (e.g., lines: 179-189). I would like the authors to rephrase these sentences and make the abstract more logical.*

We agree with the reviewer that the strengths and limitations in using rainfall datasets for variability and trend analyses should be clarified and we made amendments to the manuscript to that effect. Specifically, we pointed out that in-situ rainfall data (i.e. weather station data) has limited spatial information in a complex terrain environment such as the Rio Santa basin. Merged satellite-based precipitation data on the other hand can suffer from spatio-temporal inconsistencies related to sensor overpass times, sensor changes, as well as biases introduced by the retrieval algorithm or lack in reference in-situ data. Cross-validation of trends with independent datasets hence considerably increases trend robustness. Other precipitation metrics that are less sensitive to temporal inconsistencies are likely to be less affected, among these are

co-variabilities between precipitation and vegetation as indicated in Fig.2a or rainy season onset/retreat metrics. To clarify this, we added a few sentences to the Introduction section stating why we did not primarily exploit rainfall datasets. The use of CHIRPS here is justified by the fact that CHIRPS data has been used for rainfall trend studies in other regions of the Andes before (e.g. Segura et al., 2019, Torres-Batlló & Martí-Cardona, 2020), and that it is temporally more consistent than IMERG data, where the weighting between microwave and infrared data dynamically changes over time and therefore can lead to strong (and potentially unrealistic) trends as seen in Fig.3 IMERG trend. In the revised manuscript, we clearly stated that CHIRPS data for the RSB is in fact able to capture seasonal variability in rainfall (as shown by the bucket model, i.e. Fig.5) and to some extent represents trends in rainfall coinciding with sub-seasonal trends in vegetation greenness in the observed period (as shown in the small inset plots in Fig.4i and l).

*Second, some key messages are missing in the method section.*

*i) How did the authors reconcile the NDVI, precipitation and soil moisture data with different spatial resolution? It should be noted that the topographical issues in the studied mountain areas should not be ignored;*

In fact, all analyses are based on either native resolution of the datasets or spatial averaging over the entire basin as stated in the dataset description section. The only analysis where rainfall and vegetation indices data were jointly used for sub-domain calculations is the lagged correlation, where we state that "… we compared each VI pixel with the CHIRPS pixel intersecting it by using a nearest neighbor approach." (lines 157-158). Consequently, no interpolation was applied to either of the datasets and no additional errors or uncertainties related to the topography in the region should be introduced by our analysis. We added a sentence to the associated methods section to clarify and avoid the reader misinterpreting.

*ii) The authors should provide more details related to the lagged correlation, e.g., the mathematic implementation of a cross correlation function;*

We added an explanation of the lagged correlation, along with the corresponding equations.

*iii) It is interesting that why evapotranspiration data in the "bucket" model (e.g., Eq. 1) is set as constant over the study period? In other words, the seasonal variations in soil moisture are solely determined by precipitation? Then, where is the advantage of "bucket" model against the seasonal rainfall data (e.g., Liebmann and Marengo, 2001, also Figure A1)? A better criterion to extract SOS/EOS?*

Evapotranspiration is set to a constant value for the sake of simplicity. The aim of the bucket model is to crudely represent water retention in the soil in the most basic way, which however allows to estimate plant available water rather than rainfall accumulation alone as with the methods presented in Appendix A. This allows to represent the observed asymmetry in vegetation lag at the start and the end of the rainy season, i.e. a small lag in greening after rainfall onset but a larger lag for vegetation EOS at the end of the rainy season, when rainfall ceases but soil water is still abundant. For additional explanation we added a modified version of Fig. 5 where the rainy season onset and retreat derived by the method of Liebmann & Marengo (2001) instead of the bucket model is presented. For simplicity, the KDE's represent the whole Rio Santa basin (not distinguishing between Coordillera Negra & Blanca as in the original Fig.5).

[Figure]

*iv) Similarly, the authors should state the rationale of applying two specific thresholds to define the simulated SOS/EOS (e.g., 0.2 and 0.35 m3/m3, lines: 169-172). Are these thresholds specifically optimized for the NDVI-based SOS/EOS?*

Yes, the two thresholds were determined in order to highlight the congruence between the onset and retreat of the wet season (as inferred from precipitation data and evapotranspiration estimates) and the onset/end of the vegetation period as inferred from NDVI. We acknowledge that there is a certain arbitrariness in this choice, as the two thresholds (along with the two parameters needed to define the minimum and maximum value of the soil water bucket, viz. 0.05 and 0.5 m3/m) are plausible but not measured physical properties of the soils and vegetation types occurring in the region. We would like to reiterate here that the purpose of this exercise was not to provide a realistic simulation of the soil moisture balance and its seasonal variations at the pixel level, but rather to show i) that accounting for a lag in the response of the soil-vegetation system to the onset and retreat of the rainy season can explain the seasonal patterns derived from the NDVI data aggregated at regional level, and ii) that for this purpose CHIRPS provides a reasonable representation of the precipitation regime. Similar considerations motivate our choice of setting evapotranspiration constant to 2 mm/d in this context.

We are aware that the agreement between simulated and NDVI-derived metrics of land surface phenology could be improved by using a more realistic formulation and parameterization of the soil moisture balance. To make this clearer for the reader, we introduced amendments to the corresponding sections (Introduction and section 2.6), stating the rationale for introducing the bucket model.

*Third, the authors realized the NDVI signals were lagged behind the precipitation (e.g., Figure 2.a). i) Why not presented the variations of precipitation and NDVI after few months (instead of Figure 3). In theory, it could be able to support the coherence of SOS/EOS inferred from vegetation index and precipitation data. Unfortunately, it looks like the NDVI is always greening even take the lagged months into consideration.*

We apologise if we do not correctly understand the point of improvement the reviewer is suggesting here. The coherence of NDVI/precip SOS/EOS is shown (based on entirely independent models) and discussed in Figure 5 and could not be visualised based on annual trend plots as shown in Figure 3, even if a 1 or 2 month lag was introduced to the NDVI. As described in the manuscript, the purpose of Figure 3 is exclusively to present average regional trends, and the difficulty in reconciling datasets based on this metric. We then go on to illustrate that there is however good coherence in SOS/EOS metrics between NDVI/CHIRPS and some correspondence specifically in the late rainy season trend, allowing us to identify this as a robust feature of sub-seasonal increased water availability with direct effect on plant greening.

*ii) As is shown in Figure 6, the lags between SOS derived from MODIS NDVI and CHIRPS rainfall data and that for EOS were the same?*

As the reviewer mentioned before, section 2.5 describing the lag correlation was incomplete and was updated in the new manuscript version. We agree that interpretation of the maps shown in Fig.6d) and h) may have been difficult. For explanation, the "best lag" values shown in Fig 6d) show the median of 20 years of optimized lags. These were calculated as follows: for each gridpoint, the seasonal evolution of the NDVI and precipitation were selected similar as done in Fig. 2 for all seasons for the spatially aggregated values; next, we determined by how many days the seasonal curve of one variable had to be shifted to obtain the highest Pearson's r value. The main outcome of this analysis is a qualitative representation of the effect of topographic/spatial features on the rainfall-vegetation relations (as described in lines 217-224).

---

## Author Response (AR1)

Dear editor and reviewers,

please find below our point-by-point reply to the comments made by the reviewers and the editor. Since a few questions were raised during the revision process of the manuscript which we discussed with the editor through email exchange, we have added this correspondence down below for the sake of transparency. References to line numbers refer to the revised manuscript including track changes.

**Editor comments:**

*Dear authors*

*Thank you for submitting your responses to the reviewers' comments. I found Reviewer 2's comments to be very insightful. I am basing my decision on Reviewer 2's comments and my own reading of the paper. The topic of the paper is certainly very interesting. You use science to address perception, which is a huge challenge. Also your approach could be useful to study data-poor regions. However, some of your methodological choices are simplistic. Perhaps because of these issues, both reviewers gave mediocre ratings to this paper in our internal rating system. Also your responses to Reviewer 2's comments on these issues should have been more rigorous. I believe the paper has great potential and that is why, instead of rejecting it, I am giving you an opportunity to fix it. I am recommending major revision and your submission will be reviewed again. In your revised manuscript I request you to pay extra attention to points (i), (iii), and (iv) raised by Reviewer 2.*

Thank you for your helpful comments and giving us the opportunity to address the raised points by reviewers and within further email exchange. We detail the implemented changes in the revised manuscript based on your comments below.

*In (i), I am concerned that 0.05 degree (~ 5 km) grid spacing may not be sufficient to resolve the topography of the region.*

We agree that the lagged correlation may have been subject to uncertainties induced by the usage of CHIRPS dataset on pixel scale. In the revised manuscript we removed this analysis. Please refer to our answer to Reviewer #2 for details.

*In (iii), I don't understand why you have picked a bucket model, that too with a constant ET rate. Because your entire analysis is based on water availability, a sophisticated hydrology model capable of simulating ET and run-off is the appropriate tool. If you don't want to go that far, at least you should conduct sensitivity studies with different values of ET to quantitively evaluate your choice.*

We have decided to remove the bucket model from the manuscript as it was not key to the paper and to reset the focus of the paper on NDVI and water availability. After considering implementing a dedicated hydrological model, we concluded that using such a model would be subject to further uncertainties given by the same resolution constraints of available rainfall data the Editor and Reviewer #2 were concerned about. We now emphasize that NDVI reflects plant available water rather than sub-seasonal rainfall distributions but we do not attempt to specify how rainfall is redistributed within the valley as this is neither the focus of our study. Instead, we discuss regional-average NDVI sensitivity to monthly and annual rainfall anomalies.

*In (iv), please examine the robustness of your thresholds by conducting sensitivity studies using a range of values.*

Not done due to removal of the bucket model.

*I am looking forward to your revised manuscript.*

*Sincerely*

*Somnath*

**Author's email response:**

Dear Somnath,

First of all, thank you for giving us the opportunity to revise our paper and taking the time. In view of the points Reviewer #2 raised and you further emphasized, we would need some additional clarification on the revision points. We very much regret that we were evidently not successful in conveying the key points of our study, which we think raised some expectations that this study is not intended to meet, e.g.

R#2: "Moreover, the authors used gridded precipitation data as a proxy of water availability (e.g., lines: 179-189)" and from your e-mail: "Because your entire analysis is based on water availability, a sophisticated hydrology model capable of simulating ET and run-off is the appropriate tool."

However, our manuscript does not constitute a model-based hydrological study. Rather, we argue that based on the fact that NDVI is significantly rainfall-sensitive in the study region, it can serve as a highly-resolved and temporally-consistent indicator for changes in plant available water at sub-catchment scale (with linked implications for agriculture). Interpreting NDVI changes as changes in rain-controlled plant available water overcomes the rainfall-data resolution constraint (Rev2 point (i)) as well as the limited trustworthiness of trends in gridded rainfall products at the catchment scale – an approach that has potential for transferability to other semi-arid valleys in the Andes.

The sole purpose of the simplistic bucket model in this context is to establish precipitation as the dominant driver of regional-average NDVI variability (represented by SOS/EOS co-variability). Instead, we now intend to illustrate this relationship based on rainfall-explained variance of monthly regional-average NDVI anomalies (new panel which will be integrated into Fig2, see below), rendering the bucket model redundant. We would hence suggest to remove the bucket model along with related rainfall/NDVI lag analyses (ξ in Fig5 and Fig6d; no other figures affected), as the model raises process-based hydrological questions it was not designed to answer, that deserve evaluation beyond the scope of this paper, and that are not associated with our key findings.

Unfortunately, most discussion points revolved around the bucket model, whose application we think was misunderstood to be key to this study. Therefore and in light of the above, we would be grateful for feedback on whether you would consider the removal of the bucket model to be an acceptable step to take for the revision.

Yours sincerely,

Lorenz and co-authors

**Editor's email response:**

*Hi Stefan*

*Thanks for your email and apologies for the delayed response. The bucket model became the main issue in the manuscript because I and the reviewer feel that accurate representation of surface hydrology is crucial in your study. NDVI can change due to water availability.*

Correct. After removing the bucket model, we now show this NDVI sensitivity based on co-variability with soil moisture in Fig. 2a. This sensitivity to water availability / soil moisture and the linked suitability of NDVI as an indicator for water availability changes is the main focus of this study, which we clarified throughout the manuscript.

*That water can come from local precipitation or subsurface transport. When you say that rainfall is the dominant driver, that could just be a coincidence.*

The discussion around this point seems to be linked to imprecise language from our side. Rather than referring to local rainfall, "rainfall as dominant driver", we referred to total annual rainfall averaged across the valley, which will locally be redistributed via subsurface flow according to topography, soil types etc. We did not intend to exclude such redistribution processes from the explanation for localized greening patterns.

However, the dry-season greening patterns (in Fig. 4) that dominate the identified trend (Fig. 3a) are not highly localized but widespread throughout the entire valley. This includes the Cordillera Negra, where both glacier runoff and multiannual water storage is absent. This strongly suggests a meteorology-driven increase in dry-season soil moisture encompassing the entire RSB, which in turn drives dry-season NDVI greening at that time (l.356-361). We also acknowledge that NDVI is integrating spatial redistribution of rainfall through hydrological sub-surface processes (i.e. l. 95-97). In this context, we now illustrate the strong co-variability of NDVI with soil moisture anomalies in Fig. 2a, while inter-annual rainfall sum variability explains more than 50% of valley-average NDVI anomalies. Since we do not provide an explicit rainfall attribution however, we re-emphasized the study focus on plant water availability and only discussed possible rainfall relationships in view of existing co-variability between NDVI and rainfall.

*We need a rigorous way to test this hypothesis. I think we can answer this question using a sophisticated hydrology model that simulates subsurface transport. But this is only my opinion. Perhaps you can come up with an innovative methodology that does not include hydrological models. so I have no issues if you want to drop the hydrology model bit. But the bottom line is that your paper must provide conclusive evidence that rainfall is the dominant driver of NDVI patterns.*

We clarified the focus of this study on changes in plant available water. Our aim is not to attribute local NDVI trends to local rainfall versus other hydrological parameters such as sub-surface flow.

*If successful, this paper can be very valuable to study data poor regions.*

*Sincerely*

*Somnath*

**Reviewer #1:**

*Overall, the manuscript is well written, and has several interesting findings.*

We thank the reviewer for the evaluation and the comments.

*But my concern is that its title "Vegetation indices as proxies for spatio-temporal variations in water availability in the Rio Santa valley (Peruvian Andes)" is inadequate to what they are presenting on the paper. Based on the title, I expected the manuscript will be more focusing on the technical issues of how well satellite-based vegetation index captures spatiotemporal variations in water availability. However, the manuscript provides general characterizations about the relationships between vegetation index and precipitations, land surface phenology*

*retrieval, and land surface phenology and larger-scale circulation patterns (i.e., ENSO).*

We agree that the old title might have suggested a different focus, therefore the revised manuscript is titled: "Widespread greening suggests increased dry-season plant water availability in the Rio Santa basin (Peruvian Andes)"

*They also presented long-term greening and browsing without specific attributions of why. Therefore, I would recommend revising the title and relevant sections and expressions, especially for their overall goals.*

We updated the manuscript and changed relevant sections and expressions and specifically discussed which mechanisms drive (and which do not drive) the observed greening patterns (e.g. l. 19-21, 238-239, 315-320). Additionally we reformulated in a more specific way our goals (l.104-113), adopting a 1:1 correspondence to the subsections in Section 3, "Results".

**Review #2:**

*In this analysis, the authors attempted to provide a new alternative to track the spatial-temporal variations in water availability in Rio Santa basin (RSB). They acknowledged the limitation of in situ and remote sensing-based rainfall datasets for complex terrain, i.e., the problematic quality and temporal consistence. While it is surprising that the authors found that satellite-derived vegetation greenness (also phenology) was coupled well with recent changes in rainfall (CHIRPS, a combination of satellite and rain gauge data). The authors proposed a "bucket" model to better fit vegetation phenology derived from MODIS NDVI, the concerns also raised in the criteria in extracting SOS/EOS. Some additional serious issues related to the methods and result interpretation, the organization of discussion would weaken the reliability and implication of this study. A major revision is therefore recommended.*

We thank the reviewer for the evaluation of our manuscript and the insightful comments.

*First, I have noticed that the authors claimed that both in situ and remote sensing precipitation datasets were questionable in signifying the changes in the timing and intensity of the wet season (e.g., lines: 4-7). However, it is quite strange that the authors used such dataset to demonstrate that the pattern of precipitation*

*occurrence and the seasonality of vegetation indices are tightly coupled (e.g., lines: 10-11).*

We agree with the reviewer that the strengths and limitations in using rainfall datasets for variability and trend analyses should be clarified and we made amendments to the manuscript to that effect. In particular, gridded rainfall data uncertainty is likely to decrease with spatio-temporal aggregation (as random errors cancel each other out). Similarly, inter-annual variability (whether a year is drier or wetter than usual) is more likely to be captured by rainfall datasets than quantitative multi-decadal trends. We thus focus on rainfall comparisons for valley-average rainfall in the revised manuscript, and trends are evaluated for several rainfall datasets to estimate their uncertainty range. We do not attempt to exploit any rainfall dataset for information on small-scale, local sub-valley rainfall variability due to stated dataset limitations.

For the valley-average, vegetation seasonality is indeed tightly coupled to accumulating rainfall over the rainy season, as illustrated by Fig. 2c. Furthermore, updated Fig. 2b illustrates that inter-annual variability of regionally aggregated CHIRPS rainfall anomalies correlates with the variability in NDVI anomalies, with 52% explained variance. We updated the lines mentioned by the reviewer, clearly pointing out that the analyzed rainfall is of limited value to understand changes in *local* water availability but can give information on average seasonal water input on the regional scale.

*Moreover, the authors used gridded precipitation data as a proxy of water availability (e.g., lines: 179-189). I would like the authors to rephrase these sentences and make the abstract more logical.*

This is a very important point that did not become sufficiently clear in the previous manuscript version and is now emphasized: we do not use precipitation as a proxy for local water availability to plants. Instead, our focus is on high-resolution NDVI as an integrated proxy for local plant available water, reflecting the local effect of valley-average rainfall and subsequent water redistribution via surface runoff and subsurface flow. We now also included SMAP soil moisture estimates to illustrate the significant monthly co-variability between NDVI and soil moisture on the regional scale in Fig. 2a. On the other hand, we do explore the role of rainfall on controlling NDVI on the regional scale. While the monthly correlation with rainfall is low (Fig. 2a), this considerably increases for

annual rainfall anomalies (Fig. 2b), illustrating the control of seasonally accumulated rainfall on annual NDVI anomalies.

*Second, some key messages are missing in the method section.*

*i) How did the authors reconcile the NDVI, precipitation and soil moisture data with different spatial resolution? It should be noted that the topographical issues in the studied mountain areas should not be ignored;*

In fact, all analyses are based on either native resolution in case of the NDVI (250 m) dataset or spatial averaging was performed over the entire area of interest (Rio Santa basin), as stated in the dataset description section. The only analysis where rainfall and vegetation indices data were jointly used for sub-domain calculations was the lagged correlation. As the reviewer pointed out, potential uncertainties through topographic issues induced into the analysis by using CHIRPS at pixel scale were detrimental to the legitimacy of our study. For this reason and because the results associated with the lagged correlation were not the key messages in our paper, we removed this lag correlation analysis from the revised manuscript. Consequently, all analyses of other datasets than MODIS NDVI are conducted using only averaged values over the study domain.

*ii) The authors should provide more details related to the lagged correlation, e.g., the mathematic implementation of a cross correlation function;*

As stated in the previous point, the lagged correlation analysis was removed due to limited robustness of pixel-scale CHIRPS analyses and to stress the focus of the manuscript on the NDVI-based main messages of the study.

*iii) It is interesting that why evapotranspiration data in the "bucket" model (e.g., Eq. 1) is set as constant over the study period? In other words, the seasonal variations in soil moisture are solely determined by precipitation? Then, where is the advantage of "bucket" model against the seasonal rainfall data (e.g., Liebmann and Marengo, 2001, also Figure A1)? A better criterion to extract SOS/EOS?*

Joint answer to points iii) and iv) below.

*iv) Similarly, the authors should state the rationale of applying two specific thresholds to define the simulated SOS/EOS (e.g., 0.2 and 0.35 m3/m3, lines: 169-172). Are these thresholds specifically optimized for the NDVI-based SOS/EOS?*

After consultation with the editor, we decided to remove the bucket model as it was not key to this study. We modified the manuscript to emphasize the focus on our NDVI-based results on changes in plant available water, while discussing possible rainfall controls in that context. Hence, no SOS/EOS metrics are derived based on rainfall data. Instead, we now illustrate strong monthly co-variability between NDVI and soil moisture in the new Fig. 2a, highlighting NDVI sensitivity to available water. One of the constraints of our first manuscript was that we evidently did not clearly state that we distinguish between interannual variability in precipitation and sub-seasonal variability. Using the bucket model, we wanted to explore and test if and how NDVI variations are affected by these rainfall variabilities and whether NDVI variability is mainly governed by variability found in rainfall datasets. We did not intend to conduct a hydrological study on intraseasonal distribution of rainfall inputs into the hydrological system. Therefore, we clarified that our main focus is changes in plant water availability in the context of local agriculture and made the following amendments to that effect:

- l. 9-10: rewording: instead of "hydrological changes", "changes in plant available water"
- l. 11-15: here we stated that we will "[...] reveal a robust and highly resolved picture of recent changes in rainfall and vegetation phenology [...]". We acknowledge this formulation was misleading and might have led to misinterpretation. Reworded.
- L.53-54, 75, 81-82, 180-181, 291, 294-295, 371, 407, 416: rewording: usage of the term "water availability" and/or acknowledging influences of other components of the hydrological cycle instead of solely using "rainy season" or "rainfall" or similar.
- L.297-302: It remains unclear whether inter-annual legacy effects on water availability imposed by strong/multi-year El Niño exist or if these are a coincidence. In the context of the discussion about water distribution in the hydrological cycle, we added an argument on the possibility of a critical role of larger water storage terms into the Discussion.

Additionally, we emphasized the added value of employing vegetation-based analysis over pure rainfall dataset exploitations in this context by making the following amendments:

- l. 315-326: In this revised paragraph, we discuss the agreement of monthly CHIRPS and NDVI trends, stating that CHIRPS rainfall data appears to be appropriate for certain analysis in the Andes but we stress that results need to be compared with independent data, as for example shown by coinciding dry season NDVI and precipitation trends in Fig. 4. Furthermore, we state that our analysis is valuable as it sheds light on questions raised by analysis of inconclusive rainfall data conducted by other authors.
- l. 330-334: similar point but additionally to our illustration of the issues with the precipitation we emphasize the rationale of our analysis.

The foundation of our work is that the available rainfall datasets are not able to address the issue of changes in very local plant water availability. Datasets such as soil moisture are not available for near-climatological timescales such as the MODIS era although they probably are more suitable to understand ongoing changes in water availability than rainfall datasets alone. Therefore, we exploit MODIS NDVI as in the particular semi-arid climate, changes in plant water availability can be deduced. This includes all hydrological processes that can lead to plant greening. In the revised manuscript, we introduced SMAP soil moisture observations, which show better correlations with sub-seasonal rainfalls than any rainfall data (compare revised Fig.2). Unfortunately, SMAP (and/or other comparable) data on soil moisture is not available in an adequate spatio-temporal resolution (9km resolution, only since 2015) which is the reason why we did not incorporate such data initially. We added a new paragraph (l. 90-100) stating the rationale behind exploiting NDVI and how it relates to the components of the hydrological cycle.

*Third, the authors realized the NDVI signals were lagged behind the precipitation (e.g., Figure 2.a). i) Why not presented the variations of precipitation and NDVI after few months (instead of Figure 3). In theory, it could be able to support the coherence of SOS/EOS inferred from vegetation index and precipitation data. Unfortunately, it looks like the NDVI is always greening even take the lagged months into consideration.*

We apologize if we do not correctly understand the point of improvement the reviewer is suggesting here. The overall goal of our study is to explore whether there were changes in local plant available water in the Rio Santa Basin over the last 20 years (as suggested by Fig.3a). We are not aiming to establish robust

rainfall-NDVI lag behavior on several time scales, which is why lag analyses were removed from the revised manuscript version, clarifying the study focus.

However, we do know that NDVI shows a lag to rainfall, particularly early in the rainy season when biomass is built up. We hence include a 1-month lag in the updated Fig. 2a between CHIRPS rainfall and MODIS NDVI which slightly improves the correlation between the datasets, but just serves to quantify monthly co-variability.

In line with the reviewer suggestion, we now also show inter-annual co-variability between CHIRPS and NDVI in Fig2b, showing that there is indeed a strong link between annually aggregated anomalies of detrended rainfall and plant greenness (52% explained variability), with drops in NDVI during years with low rainfall sums. This establishes annual rainfall as an important factor for annual NDVI variability on the regional scale. The reviewer is correct however that the NDVI trend identified in Fig. 3 is not reflected in the rainfall dataset trends, which can have various reasons. Apart from rainfall dataset uncertainties, non-linear vegetation sensitivities to rainfall on the seasonal time scale may play a role for NDVI trends that do not reflect in annual rainfall totals (discussed in the revised manuscript in l. 369-373). For example, while we do not identify annual trends in any of the rainfall datasets (Fig. 3), we do find indications for increased CHIRPS rainfall in the early and during the dry season, coinciding with strongest NDVI trends (Fig.4). Such dry-season rainfall trends contribute little to annual rainfall trends in absolute terms, but may help vegetation to delay senescence towards the end of the season and throughout the dry season, thus disproportionately affecting annual-average greening.

Regarding greening seasonality, the reviewer is correct that the NDVI shows some greening signal throughout the season but the strongest greening is visible throughout the dry season (compare e.g. the pie charts in Fig.4 giving the relative amount of greening pixel compared to the total available pixels) which suggests changes in water availability as the main driver in line with high NDVI sensitivity to soil moisture (Fig 2a). Furthermore, we would like to point out that during the months Oct-Feb only a very limited amount of pixels show greening and we suspect that some percentages of these pixels can be attributed to land-cover change as we discussed in l.344-348. In this context, we also added one sentence (l.411-412) justifying the attribution of the dry-season greening trends to increased water availability as compared to greening driven by $CO_2$ fertilization, as

the latter should be especially visible when water stress is low (i.e. during the rainy season).

*ii) As is shown in Figure 6, the lags between SOS derived from MODIS NDVI and CHIRPS rainfall data and that for EOS were the same?*

In the revised manuscript the lag correlation was removed as stated before. As we mentioned during the discussion, our methodology on lag correlation did not incorporate sub-seasonal lag analysis and would only have been optimized for each growing season (from 01-09 to 31-08 of each year). Independent of the removed methodology, we would like to take the opportunity to clarify: The reviewer is correct, the temporal lags at SOS and EOS are not expected to be the same. In fact, the revised Fig. 2c is conceptualizing the principle in a systematic way. With the start of seasonal rainfalls, soil moisture almost immediately responds while after the retreat of the seasonal rains there still remains moisture in the system creating a larger lag towards the end of the season. NDVI is lagged even further behind the soil moisture signal as biomass can obviously not be accumulated instantaneously at the start of the season. At the end of the season - generally spoken (in reality of course dependent on plant species) - plants tend to react to decreasing water availability by employing conservative water-use strategies which consequently leads to a delayed response of NDVI in comparison to the hydrological parameters.

---

## Author Response (AR2)

Dear editor and reviewer,

please find below our point-by-point reply to the comments made by the reviewer. References to line numbers refer to the revised manuscript including track changes.

**Reviewer #2 comments:**
After discarding those controversial or unclear analyses in the original submission, the revised MS has been greatly improved. Most of my following comments are editorial.

We thank the reviewer for the evaluation of our manuscript, the insightful comments and the reviewer's interest in our manuscript.

Line 26: change "increasingly" to "significantly".

Changed.

Line 101: the now deleted words state water availability in the region does not match time series of rain-gauge data. I am curious why rain gauge data is not correct?

The deleted sentence refers to the previously published analyses of weather station data in the "context of perceived changes by local farmers". As stated in the Introduction section, Gurgiser et al. (2016) found in interviews that farmers report detrimental effects to their agriculture in the past decades. Their analysis of available rain-gauge data however did not show any meteorological trends or an increase in extreme events as reported by the farmers. The argument we originally wanted to make here (and still do in the revised manuscript from l. 43 and discussed from l.301) is that to date it remains unclear whether the mismatch of farmers' reports is due to incorrect rain gauge data or their perception not meeting actual changes (or not-changes).

Line 171: Are there any references to the extraction of SOS/EOS, especially the threshold of "30%"? rather than the generally used 50%?

We used 30 % as this is the default setting in the DATimeS software (Belda et al., 2020). We reformulated the sentence to clarify (see l. 150).

Line 176: How do you define "POS"?

Good point, in contrast to SOS and EOS it is undefined in the current version. POS (Peak Of Season) is defined as the day of season where the maximum NDVI value occurs. Added to the revised manuscript (see l. 150).

Line 181: suggest to delete "variability in"

Deleted.

Section 3.1: I did not quite understand the rationale of introducing soil moisture data here. (1) Soil moisture agrees better with NDVI in monthly scale, and such a advantage is not clear at annual scale. (2) There is no sufficient proof showing precipitation is a better indicator than soil moisture. Then, why soil moisture is not included in the following analysis?

The rationale of introducing soil moisture (SM) data was to show the relationship between NDVI and different components of the hydrology following the reviewers' comment iii) and iv) and the discussion with the editor.

Regarding (1): Yes, SM data shows higher agreement with NDVI than rainfall on monthly scale (Fig 2a). We did not perform the analysis for the annual scale (Fig 2b) as the SM data is unfortunately only available since 2015 and we do not consider correlations based on five data points to be sufficiently robust.

Regarding (2): We do not argue that precipitation is a better indicator than SM, in fact we state the opposite (from l. 173). SM is not included in the subsequent analysis as we focus on the near-climatological MODIS era 2000-2020 and the SMAP data is only available from 2015 onwards.

Line 221-222: This statement may be unfair; why soil moisture data is not presented in Fig. 2b?

See also the previous answer. On a monthly basis for 2015-2020, SM yields higher explained variance than CHIRPS but on an annual scale 2000-2020 (where comparison with SMAP is not applicable) CHIRPS reaches a relatively high explained variance.

Line 223: Fig 2b should be Fig 2d??

Correct, changed.

3.2 subtitle: and "their consistencies?

Changed.

Line 239: any "significant" changes? no changes != no significant changes

Added.

Line 253-256: The KDEs in Fig. 5 is inferred from two specific regions (e.g., Cordillera Negra and Blanca), respectively. (1) RSB is the combination of Cordillera Negra and Blanca? (2) To be straightforward, please provide the KDE map of phenology over RSB during the study year.

Regarding (1): Yes both regions together are the complete RSB watershed. In the manuscript (l. 100) we state where the eastern (Blanca) and western (Negra) parts of the RSB watershed are located and it is shown in Fig.1. However, we agree that this might be confusing.

Regarding (2): We added a direct reference to Fig.1 in the caption of Fig.5. to clarify.

Line 290: Please clarify the "mean seasonal precipitation" here. Does it refer to the averaged precipitation during the period 2000-2020 or the neutral years??

Reformulated. It refers to the sum of the mean monthly precipitation sums for the period 2000-2020 and not to the neutral years.

Line 322: NDVI changes: Note that both NDVI and CHIRPS precipitation data are detrended there. NDVI changes should be NDVI variabilities?

Yes, agreed and changed.

Line 343-345: I do not suggest to add these sentences (from "they cannot explain" to "land-cover change." here, since the authors have excluded regions with land-cover changes, see lines: 188-191

To clarify, we think that we rather accounted for land-use effects (such as irrigation which decouples NDVI and water availability) but we did not account for land-cover change (l. 168: "We did not account for land-cover change during….."). We therefore would prefer to keep these statements as we believe our analysis is more reputable if we discuss potential weaknesses and limitations.

Line 377: We cannot find a significant advance in SOS ...SOS in RSB is featured with "high interannual variability but no significant trend"

Reformulated.

Line 394: "water availability" should be "plant water availability" ?

Added.

Fig.1: (1) Why these cities (or sites) are not included in the main figure? (2). There is no obvious East-West gradient in precipitation among these cities (sites).

Regarding (1): The reason is simply the map scale and aesthetics. On the scale of the full RSB the locations of the observations would be hard to distinguish.

Regarding (2): We agree, that pattern might be relatively hard to identify in that plot. In fact, we find the highest seasonal precipitation in the easternmost plot (Carachuco), while Llupa and Paquishca show a very similar amount (they are in sight of each other). Similarly, the two western rain gauges show a very similar signal while the rain-gauge Universidad, located at the valley floor has the lowest values which might be an altitudinal effect. To be more precise, we changed the plot from monthly precipitation sums to cumulative precipitation sums (see below for a large version) and reformulated the caption to "Lower right panel roughly illustrates the East-West precipitation gradient…". Although the data shows not a strict East-West gradient at all stations, it captures that rainfall sums at the eastern Coordillera Blanca range are roughly 1000mm/season while the Coordillera Negra rain gauges receive approximately 300mm less per season.

[Figure]

Fig.2: (1). Better correlation between NDVI and soil moisture is found there. (2) why soil moisture is not included in Figure 2b? Then, it is unfair to say precipitation agrees better with NDVI. (3) Why the correlation between precipitation and NDVI differs across different timescale? Time lag effect??

Regarding (1): Correct.

Regarding (2): See our previous reply to the same question above.

Regarding (3): On a monthly basis, the correlation of NDVI and precipitation is comparably low. In Fig. 2a this correlation is based on the data of 2015-2020 due to the availability of the SM data. It yields a coefficient of determination of 0.11. For 2000-2020 this value is slightly improved but still similarly low (see plot below). In contrast, on an annual scale this correlation drastically improves. The reviewer is correct that this is related to a time-lag effect but we would rather describe it as NDVI being a cumulative indicator of water availability which in turn is governed by inputs into the hydrological system (i.e. precipitation). Consequently, anomalously wet seasons yield anomalously green vegetation and vice versa. The near-immediate response as interpreted through monthly anomalies is less clear for several reasons: 1) in the wet months we do not expect a direct response of vegetation to additional rainfall inputs but this additional moisture might be beneficial after the retreat of the seasonal rainfalls, 2) similarly, this might be the case for the driest time of the year and 3) the time lag between NDVI and precipitation is varying dynamically over the season as schematically presented in Fig. 2c.

[Figure]

$R^2 = 0.139$, P 0.0, detrended, monthly anomalies, ndvi shifted -1 month

Fig.3: How about providing a map showing the day season changes in NDVI and precipitations?

We apologise that we do not understand what the reviewer means with "day season changes in NDVI and precipitations".

Fig.4: I do not think it is wise to only take pixels with significant changes in NDVI into consideration. For example, the average NDVI trends, Mar (4g) vs. May (4i).

We are not completely sure what the reviewer is suggesting here. The data availability differs between the rainy (e.g. March) and the drier months (e.g. May). The idea about showing these maps is to transparently show where and when trends occur and data availability issues due to cloud cover (grey in the map) during the rainy season. We do not use average monthly trends of NDVI in any plot or sentence throughout the manuscript. In case the reviewer is referring to the inset plots: These are monthly domain average CHIRPS data as stated in the caption: "Small panels show domain mean CHIRPS rainfall data for the respective month and additionally decadal slope (m) and linear regression statistics for significant ($p < 0.05$)". We added a few words to the figure caption to clarify.

Fig.6: LOS is mainly determined by EOS

We believe the reviewer mixed up SOS and EOS. We mentioned in the text (l 205) that LOS is mainly governed by SOS. It has a high variability over the period of observation with a variability range of up to 60 days while EOS only varies up to 34 days (compare e.g. Section 3.3 or Fig. 5). We added a sentence in the text (l. 212), explaining the effect of the EOS trend on LOS in the context of the SOS variability.

---

## Author Response (AR3)

Dear Somnath,

please find below our reply to the comments. References to line numbers refer to the revised manuscript including track changes.

1. "Section 3.1: I did not quite understand the rationale of introducing soil moisture data here...."

We have updated the manuscript clarifying the rationale behind introducing soil moisture. (from l.172). The availability of soil moisture is what drives plant growth/senescence and thus we observe a good correlation with NDVI at monthly time scales, and a poor one with precipitation. The SMAP soil moisture record however is too short to allow for a meaningful analysis of interannual and decadal variability, which is why for that purpose we exploit the longer precipitation records, which at these time scales works well.

2. "Fig.2: ... (2) why soil moisture is not included in Figure 2b? Then, it is unfair to say precipitation agrees better with NDVI.

We updated Fig. 2 by including soil moisture into panel 2b.

(3) Why the correlation between precipitation and NDVI differs across different timescale? Time lag effect??"

We have added a few sentences explaining why the correlation between precipitation and NDVI differs across time scales (from l. 258).

I am OK with your responses but I think these clarifications should also be in the manuscript. The readers may also have the same questions. Kindly add brief 2-3 sentence clarifications at appropriate locations and resubmit the manuscript.

We hope that with our reply above and the corresponding changes in the text, our manuscript can now be accepted for publication.

Kind regards,

The authors